# Gut Hormones as Potential Therapeutic Targets or Biomarkers of Response in Depression: The Case of Motilin

**DOI:** 10.3390/life11090892

**Published:** 2021-08-29

**Authors:** Ravi Philip Rajkumar

**Affiliations:** Department of Psychiatry, Jawaharlal Institute of Postgraduate Medical Education and Research, Pondicherry 605006, India; ravi.psych@gmail.com; Tel.: +91-413-229-6280

**Keywords:** depression, gut–brain axis, motilin, serotonin, gamma–aminobutyric acid, gonadal hormones, hypothalamic–pituitary–thyroid axis, antidepressants, macrolide antibiotics

## Abstract

Recent research has identified the gut–brain axis as a key mechanistic pathway and potential therapeutic target in depression. In this paper, the potential role of gut hormones as potential treatments or predictors of response in depression is examined, with specific reference to the peptide hormone motilin. This possibility is explored through two methods: (1) a conceptual review of the possible links between motilin and depression, including evidence from animal and human research as well as clinical trials, based on a literature search of three scientific databases, and (2) an analysis of the relationship between a functional polymorphism (rs2281820) of the motilin (MLN) gene and cross-national variations in the prevalence of depression based on allele frequency data after correction for potential confounders. It was observed that (1) there are several plausible mechanisms, including interactions with diet, monoamine, and neuroendocrine pathways, to suggest that motilin may be relevant to the pathophysiology and treatment of depression, and (2) there was a significant correlation between rs2281820 allele frequencies and the prevalence of depression after correcting for multiple confounding factors. These results suggest that further evaluation of the utility of motilin and related gut peptides as markers of antidepressant response is required and that these molecular pathways represent potential future mechanisms for antidepressant drug development.

## 1. Introduction

Depression is one of the most common mental disorders, affecting over 25 million individuals worldwide, with an estimated 12-month prevalence of 5.5–6% and a lifetime prevalence of 11–15% across various countries [1,2]. Depression is estimated to be one of the leading causes of disability; at a global level, it is the fourth leading cause of disability in adolescents and the sixth in adults [3]. Though effective treatments for depression have been developed, the efficacy of pharmacological treatments is modest in many patients [4], with only half of the patients showing a significant response and one third responding completely to a given drug [5]. Aside from that, these medications are frequently associated with troublesome adverse effects, particularly in certain age groups [6,7]. Though several psychological interventions have an efficacy comparable to that of medications, unsatisfactory responses are often observed, and these treatments may not be easy to deliver or access in low- or middle-income countries [8,9]. An important limitation of existing antidepressant medications is that they are largely based on the monoamine hypothesis of depression, in which depressive symptoms are considered to arise from dysregulation of the brain’s noradrenergic, serotonergic and dopaminergic pathways [10]. More recently, medications acting on both monoamine and melatonin receptors have been approved for the treatment of depression, and though effective, they do not appear to offer a significant advantage over older drugs in terms of efficacy [11]. Contemporary models of depression, while recognizing the importance of these mechanisms, have identified a variety of other molecular and cellular processes involved in the pathogenesis of this disorder. These include changes in the neuroendocrine and stress-related pathways, alterations in neural plasticity and neurogenesis, activation of immune-inflammatory pathways and the involvement of transmitters other than monoamines, such as glutamate, gamma–aminobutyric acid (GABA) and neuropeptides [12,13,14,15]. It is now increasingly recognized that further advancements in the treatment of depression are likely to arise from research targeting these pathways and mechanisms [16]. The efficacy of novel antidepressants such as ketamine, which targets glutamatergic receptors, and brexanolone, a progesterone derivative acting through GABA receptors, provides some support for this perspective [17,18].

In recent years, data from several converging lines of research have brought the role of the gut–brain axis in the pathophysiology and treatment of depression into focus. Research in animal models as well as human subjects has shown that gut–brain interactions, involving the gut microbiome as well as neural and endocrine mechanisms, are involved in modulating emotions and social behavior [19]. Stress can alter the composition of the gut microbiome, while modification of the microbiome may enhance resilience through stress via gut–brain signaling [20]. Communication between the gut and brain is mediated through several mechanisms, including afferent signals transmitted through the vagus nerve, changes in immune-inflammatory activity and changes in the metabolism of specific amino acids and fatty acids [21]. Evidence of interactions between stress and gut microbiota, leading to changes in intestinal permeability and immune-inflammatory activation, have been documented in patients with depression [22]. A recent analysis of over 3000 patients with major depressive disorder found that over 70% of patients reported at least one gastrointestinal symptom. In these patients, the severity of these symptoms was significantly associated with suicidal ideation and attempts [23]. Investigations of the mechanisms linking the brain and gut in patients with depression have largely focused on changes in the gut microbiome, particularly in response to pharmacological treatment in both animal models and human subjects [24,25]. It is likely that several biological pathways mediate the link between changes in the gut microbiota and depressive symptomatology, including adrenal and gonadal steroid hormones and cytokines [26].

More recently, it has been proposed that the group of hormones known as gut peptides or gut hormones may play an important role in brain–gut signaling in this disorder. These hormones are a heterogeneous group of peptide hormones whose primary function is to regulate various aspects of gastrointestinal function. However, receptors for these hormones have also been discovered in various other sites, such as the central and autonomic nervous system and immune cells [27]. There is preliminary evidence that some of these hormones, such as ghrelin, neuropeptide Y, glucagon-like peptide 1 and cholecystokinin, are related to the stress response and the emergence of depressive symptomatology in humans [28]. Among these hormones, ghrelin has received the most attention from researchers to date. This peptide hormone was initially identified as the endogenous ligand for receptors stimulating the release of growth hormone (GH), hence its “GH-releasing agent” name [29]. Subsequently, ghrelin has been noted to play an important role in the regulation of appetite and food intake, sleep, mood, cognition and responses to stress [30]. Ghrelin has been found to alleviate depressive-like behaviors caused by chronic stress in animal models [31,32]. Studies in human subjects have found elevated ghrelin levels in patients with depression, though these findings appear to be influenced by factors such as gender and the severity of depressive symptoms [33,34]. It is not clear if these findings should be taken to imply a direct causal link between ghrelin and depression or whether they represent a compensatory response in the face of chronic stress or severe depression [35]. Motilin, a 22-amino acid polypeptide hormone, has significant structural similarities to ghrelin, is co-secreted with ghrelin from specific cell types and has similar effects on growth hormone secretion and gastrointestinal motility [36,37,38]. To date, a single receptor for motilin, designated MLNR, has been identified. Activation of this receptor leads to the influx of calcium ions, primarily through L-type calcium channels [39]. Despite the similarities between motilin and ghrelin, there is little research in the literature examining a possible link between motilin and depression in humans. In this paper, several converging lines of evidence are presented to suggest that such a link is biologically plausible and that motilin may have promise both as a marker of response to existing antidepressants and as a potential therapeutic target.

## 2. Materials and Methods

The current study was carried out in two stages. In the first stage, the existing literature on the relationship between motilin and depression was reviewed through a search of the PubMed, Scopus and ProQuest databases, for which institutional access was available. Search terms included “motilin” and “motilin receptor” as well as “depression”, “major depression” or “depressive disorder”. This approach yielded only a small number of relevant citations (n = 29), with no studies examining specific links between major depression and motilin in human patients. Hence, a conceptual review of possible mechanistic links between motilin and depression was undertaken. The key areas of focus for this review were identified using existing systematic reviews of gut–brain axis interactions in depression as a guide [40,41,42]. Based on this, further searches were carried out, pairing “motilin” with the following search terms (Table 1).

The citations obtained through these searches were examined. Articles were included in this review if they described either (1) human patients with depression or related disorders, (2) animal models of depression or (3) in vivo or in vitro analyses of the interrelationships between motilin and other relevant neurotransmitters, hormones, immune mediators, neurotrophic factors or nutrients. As the evidence obtained in the first stage showed evidence of several possible biological mechanisms linking motilin and depression in humans, a further test of this hypothesis was undertaken in a second stage using allele frequency data. Though subject to certain important limitations, this method has been used to identify associations between the prevalence of depression and other genetic variants of interest, such as polymorphisms of the serotonin transporter (5-HTTLPR), monoamine oxidase A (MAO-A) and mu opioid receptor type 1 (OPRM1) genes [45,46]; these associations have subsequently been confirmed in research involving samples of patients with depression [47,48,49].

The rs2281820 C/T polymorphism of the motilin (*MLN*) gene located on the short arm of chromosome 6 was selected for analysis. This polymorphism was studied because it has been shown to have functional consequences, and it has been associated with disorders of gallbladder motility in humans [50]. Data on the frequencies of the *MLN* rs2281820 polymorphism for samples obtained from 26 distinct countries were obtained from the Allele Frequency Database, which provides free access to data on 664,708 distinct genetic polymorphisms obtained from a wide range of populations and samples [51,52]. Allele frequencies in ALFRED are expressed as proportions for each allele. For example, the data on the rs2281820 polymorphism in a sample of 1950 Estonian subjects were expressed as 0.516 for the C allele and 0.484 for the T allele. When more than one sample was available from a given country, the weighted mean of the allele frequency for all samples from that country was used for analysis.

Information on the estimated prevalence of depression for these countries was obtained from the World Health Organization’s Global Health Estimates for the year 2017 [53]. To correct for the effect of other factors influencing cross-national variations in depression [44,46,54,55], the following potential confounding factors known to be associated with such variations were included in the analysis: gross national income (expressed as dollar values using the Atlas method), distance of each country’s capital city from the equator (as a proxy marker for climate), estimated level of cultural individualism–collectivism and per capita sugar consumption (expressed as kilocalories per capita per day). Data on the gross national income were obtained from 2018 estimates by the World Bank [56]. Data on cultural individualism–collectivism were obtained from the Hofstede Institute’s data on cross-national variations in culture [57]. Information on sugar consumption was obtained through a database query from the Food and Agricultural Organization’s Food and Agriculture Statistics (FAOSTAT) database for the year 2016 [58].

The above variables were tested for normality using the Shapiro–Wilk test. Except for per capita sugar consumption and prevalence of depression, none of the study variables conformed to a Gaussian distribution (*p* < 0.05, Shapiro–Wilk test). Accordingly, these variables were converted to an approximately normal distribution using a natural logarithmic transformation.

Following transformation, the bivariate correlations between the *MLN* rs2281820 allele frequency and the prevalence of depression, as well as correlations between these variables and potential confounders, were examined. Finally, multivariate regression analyses using both the direct (“enter”) and stepwise methods were carried out to confirm whether the link between the *MLN* allele frequencies and depression remained significant after correcting for the confounding variables. Given the exploratory nature of this analysis and the low sample size, all variables significantly associated with depression at *p* < 0.1 or lower were included in the regression analyses.

## 3. Results

### 3.1. Conceptual Analysis of the Links between Motilin and Depression

#### 3.1.1. Gastrointestinal Motility

Patients with depressive episodes have high rates of co-occurring gastrointestinal symptoms, many of which, such as constipation and dyspepsia, are related to impaired gastrointestinal motility [23]. The presence of these symptoms has been associated with more severe forms of depression [59]. Studies using electrogastrography to analyze the activity of gastric smooth muscle have found evidence of gastric dysrhythmia in patients with depression; in these patients, there was also evidence of an association between this abnormality and symptoms suggestive of sympathetic activity [60,61]. In contrast, low concentrations of motilin stimulate gastrointestinal motility through the facilitation of cholinergic activity [62], suggesting a contrast or even an antagonism of sorts between the two processes.

Functional gastrointestinal disorders, such as functional dyspepsia (FD) and irritable bowel syndrome (IBS), are associated with significantly high rates of comorbid depression, estimated at 20.9% in FD and 23.3% in IBS [63,64], representing an approximately 1.5 to 2-fold increase over the prevalence of depression in the general population [65]. In FD, which is characterized by specific symptoms of gastric dysmotility, the rates of comorbid depression increase threefold in patients with more severe symptoms [64]. Likewise, patients with depression have a twofold risk of developing irritable bowel syndrome, particularly following an acute gastrointestinal infection [66]. Altered patterns of motilin secretion have been observed in both FD and IBS, particularly in patients with more severe symptoms [67,68,69,70]. An increase in motilin release in response to stress, in comparison with healthy volunteers, has been observed in patients with IBS [71], while higher motilin levels in patients with FD have been associated with a reduced negative effect and an increased positive effect [72]. More notably, clinical trials have found that both the antidepressant mirtazapine [73] and certain phytochemicals [74,75] improve both depressive and gastrointestinal symptoms in patients with FD and that all these pharmacological agents are associated with significant increases in plasma motilin post-treatment. Functional constipation, a functional gastrointestinal disorder usually diagnosed in children and adolescents, is also associated with elevated rates of depression [76]. Children with this disorder have reduced serum motilin levels when compared with healthy controls [77].

Though no definitive conclusions can be drawn from these results, they are consistent with the possibility that motilin is associated with gastrointestinal symptoms in depressed patients, that changes in motilin levels may be a biomarker of response in patients with comorbid FD and depression and that some antidepressants may have a significant effect on the motilin levels. However, it is not known to what extent the findings in humans may have been affected by confounding factors, such as other medications or dietary patterns. The last of these points will be discussed further under “Antidepressants” below.

#### 3.1.2. Neuroendocrine Axis Functioning

The hypothalamic–pituitary–adrenal (HPA), hypothalamic–pituitary–thyroid (HPT) and hypothalamic–pituitary–gonadal (HPG) axes have all been implicated in the pathophysiology of depression to a certain extent [14,26,43]. In addition, there is evidence of blunted growth hormone (GH) release in patients with depression [78]. There is preliminary evidence of the relationships between motilin and certain components of these neuroendocrine axes.

HPA axis: Among the hormones involved in the regulation of the HPA axis, corticotropin-releasing hormone (CRH, also known as corticotropin-releasing factor (CRF)) is considered to play a central role in several processes related to depression, including overactivity of the HPA axis and reduced neural plasticity [79]. In an animal study, direct injection of CRH into the cerebral ventricles led to suppression of motilin secretion, while intravenous administration was associated with a blunted gastric response to motilin despite normal circulating motilin levels [80]. Exposure to experimental stress in a rat model was associated with increased peripheral levels of both motilin and cortisol [81]. Though interesting, these results require confirmation in human subjects.

HPT axis: Though motilin has primarily been isolated from gastrointestinal and nervous tissue, recent research has found evidence of motilin synthesis and release in the thyroid gland, both in animals and in humans. Studies in rats have found that thyroidectomy is associated with lower motilin levels and reduced gastrointestinal motility and that electrical stimulation of the hypothalamic paraventricular nucleus (PVN) led to increased motilin release from the thyroid gland [82]. These findings are relevant because constipation is a common symptom of hypothyroidism, a condition frequently associated with depression in both its subclinical and clinical forms [43,83]. It is possible that reduced motilin levels may be associated with both of these sets of symptoms. There are no studies on the motilin levels in human subjects with hypothyroidism. On the other hand, a study on patients with hyperthyroidism found lower levels of plasma motilin compared with the controls, as well as a negative correlation between thyroxine (T4) and the motilin levels. Plasma motilin showed a tendency to normalize following treatment in these patients [84]. These results suggest that either excessive or deficient functioning of the HPT axis may be associated with alterations in the motilin levels, which may be related to the frequent comorbidity between thyroid disorders and depression [85,86].

HPG axis: Evidence for a link between motilin and HPG axis functioning has been obtained primarily from animal models and research in women. In a female rat model, both the central and peripheral administration of motilin resulted in a decrease in the levels of luteinizing hormone (LH) [87]. In contrast, the motilin levels were higher during the luteal phase of menstrual cycles [88]. This suggests that LH may increase motilin levels, while motilin may reduce LH through a feedback regulatory mechanism [87]. Alternately, both may vary in relation to the levels of progesterone, as a significant positive correlation between progesterone and motilin was found in the latter study [88]. These results are of significance given the following observations regarding depression in women: (1) the established associations between depressive symptoms and the late luteal phase of the menstrual cycle, including exacerbations of pre-existing depression as well as phasic symptoms [89,90], (2) increased rates of gastrointestinal symptoms in women with a premenstrual exacerbation of depression [90], (3) evidence that exogenous progesterone analogs can induce depression, particularly in the post-partum period [91] and (4) evidence that reduced LH or elevated follicle-stimulating hormone (FSH) levels may be associated with an increased risk of depression as well as a poorer response to antidepressants in post-partum or post-menopausal women [92,93]. Though a definitive conclusion cannot be drawn from this data, there are possible correlations between the FSH, LH, progesterone and motilin levels which may be of relevance to depression, particularly in women.

GH: Reduced GH secretion, both at rest and in response to exercise or neurotransmitter receptor agonists, has been documented in children, adolescents and adults with depression [78,94,95,96]. Motilin is a potent stimulator of growth hormone release and has been considered by some authors to be a physiological growth hormone-releasing factor (GRF) [36,97]. Though the exact significance of changes in growth hormone levels in depression is a matter of debate, it has been suggested that stress-induced changes in monoamine transmission may underlie these alterations [98], while antidepressant treatment may result in a normalization of GH release patterns [99]. There is also some evidence that the administration of ghrelin, which is similar to motilin both structurally and as a GRF, results in reductions in depressive symptoms in parallel with increases in GH levels [100]. It is plausible that effects of this sort may be discovered for motilin as well.

In summary, there is at least indirect evidence suggesting that further investigation of the interactions of motilin with endocrine parameters, particularly those involving the thyroid and gonadal axes and growth hormone release, may yield clues to a better understanding of the pathophysiology and treatment of depression.

#### 3.1.3. Stress and Stress Responses

As mentioned earlier, concurrent elevations in motilin and cortisol were observed in an animal model of acute stress [81], while elevated motilin levels were observed following exposure to an experimental stressor in patients with IBS [71]. Exposure to a stressor over a period of 10 days was also associated with increased motilin levels in a rat model, and this was associated with an increased frequency of bowel movements [101]. In contrast, a more prolonged (21 days) exposure to experimental stress in mice was associated with a significant reduction in motilin levels compared with a “non-stressed” control group. This was associated with reduced gastric emptying and intestinal propulsion [102]. These results suggest that acute and chronic stress may have differential effects on motilin levels, which may be mediated through the dysregulation in HPA axis functioning induced by exposure to a chronic stressor [103]. An alternate mechanism that may link stress and motilin is the sympathetic nervous system, as a stellate ganglion block has been found to reduce cortisol levels and increase motilin levels in patients undergoing laparoscopic surgery for colorectal cancer [104]. Finally, in an animal model of chronic stress—the “forced swimming test” (FST), which is used to model depression—rats exposed to the FST showed reduced motilin levels, which were associated with both depression-like behavioral changes and reduced gastrointestinal motility [105]. Though caution is required in applying these findings to human subjects, they suggest that depression resulting from chronic stress may be associated with reduced levels of motilin. 

#### 3.1.4. Monoamine Transmitters

There is evidence that the release of motilin is regulated by monoamine transmitters such as serotonin and norepinephrine and that the effects of motilin on gastrointestinal motility may be mediated, at least in part, through these transmitters. Dopamine appears to increase motilin secretion when infused intravenously after a meal, whereas it has the opposite effect in fasting subjects, causing a decrease in plasma motilin levels [106,107]. The blockade of D_2_ dopamine receptors by domperidone is also associated with a rapid increase in plasma motilin [108]. Serotonin (5-HT) increases the release of motilin [109], and the effects of motilin on upper gastrointestinal motility are partially mediated by the 5-HT_3_ receptors; a blockade of these receptors antagonizes the effects of motilin on gastrointestinal smooth muscle contraction [110]. While there are no studies directly examining the effect of noradrenaline on motilin secretion, this transmitter is known to reduce gastric motility [111], and an artificially induced sympathetic block is associated with an increase in motilin levels [104]. These results suggest that a complex balance may exist between monoaminergic activity and motilin release at the level of the gut; however, there is no direct evidence as of yet for such effects at a central level.

#### 3.1.5. Other Neurotransmitters

Though primarily considered a regulator of gastrointestinal motility, the motilin receptor *MLNR* has been identified at various other sites, including the vagus nerve and specific brain regions [56]. At a central level, motilin receptors have been identified on neurons of the lateral vestibular nucleus in rabbits, where they exert inhibitory effects and appear to act in an additive manner with GABA. Immunocytochemical studies also suggest that motilin and GABA may be co-localized and released together by some subsets of neurons, such as cerebellar Purkinje cells [112]. Subsequently, motilin receptors were found to be expressed at high levels in the basolateral nucleus of the mouse amygdala, a brain region involved in stress susceptibility, fear and anxiety responses. Motilin, as well as the motilin agonist erythromycin, increases GABAergic transmission in this area, resulting in reduced anxiety-like responses to stress [113]. This finding may be significant in the light of evidence of structural and functional abnormalities of the basolateral amygdala in patients with depression [114,115]. Motilin receptors have also been identified in other brain regions that have been implicated in the pathophysiology of depression [116], such as the hippocampus and hypothalamus [117,118]. Motilin receptors have also been identified in the cerebellum [119], which has been implicated in the pathophysiology of recurrent depressive episodes [120]. In all these brain regions, motilin may act to enhance GABAergic neurotransmission. Research in the last two decades has found evidence of reduced GABA levels in multiple brain regions in patients with depression [15], and it has been suggested that the activation of motilin receptors may represent a potential therapeutic target in such situations [113]. Interactions between motilin and cholinergic transmission, both locally in the gastrointestinal tract and through the vagus nerve, have also been identified in human animal models. For example, the exogenous administration of motilin in human volunteers has been associated with increased hunger mediated through the cholinergic pathways [121], and certain phytochemicals have been observed to cause concurrent increases in the acetylcholine and motilin levels [122]. Motilin has also been shown to influence the release of insulin and pancreatic polypeptide levels in dogs, and both of these effects appear to involve cholinergic signaling through the vagus nerve, which possesses motilin receptors [123]. These effects are abolished by vagotomy or anticholinergic drugs [124,125]. This mechanism may also be of relevance, as vagal signaling has been identified as a potential pathway linking the gut and brain in depression [21,40], and vagus nerve stimulation may be a useful therapeutic option in some patients with this disorder [126]. Finally, peripheral administration of erythromycin, a motilin receptor agonist, can alter the activity of several key brain regions, including the orbitofrontal cortex, limbic structures, hypothalamus, caudate nucleus and putamen [127]. Though the exact neurotransmitters mediating this response have not been identified, this finding provides further evidence of the central effects of motilin on the brain circuits involved in appetitive and hedonic behaviors, whose functioning is altered in patients with depression [116].

#### 3.1.6. Immune and Inflammatory Pathways

Relatively little research has examined the interplay between motilin and immune system functions. However, the administration of the putative anti-inflammatory cytokine interleukin-11 (IL-11) was associated with an increase in the expression of motilin messenger RNA (mRNA), as well as increased motilin levels, in a rabbit model. This effect appeared to be mediated both by a direct effect of IL-11 and an indirect effect of IL-11 on leptin release, which in turn stimulated motilin synthesis and release [128]. Though there is no evidence directly linking IL-11 to the pathogenesis of depression, changes in IL-11 expression may be a significant predictor of the response to antidepressant medications [129]. Given that certain antidepressant medications produce significant changes in motilin levels [73], further analysis of the relationship between motilin and cytokines may aid in identifying markers of antidepressant response.

#### 3.1.7. Neurotrophic Factors

There is increasing evidence that alterations in neural plasticity may represent a final common pathway through which stressful life events influence the onset and duration of depressive episodes [130]. A key molecule that may mediate this process is brain-derived neurotrophic factor (BDNF), which promotes hippocampal neurogenesis and neural plasticity. Functional polymorphisms of the BDNF gene are significantly associated with the risk of developing depression after exposure to stressors [131], and depression is associated with a modest but significant decrease in serum BDNF levels compared with healthy controls [132]. In an animal model of depression using the forced swimming test, rats exposed to this chronic stressor showed significant decreases in the plasma levels of both BDNF and motilin [105], which occurred in parallel with increases in CRH and cortisol. On the other hand, BDNF levels may be increased by the activation of GABAergic pathways [133], which is one of the key central effects of motilin. These indirect links between motilin and BDNF require examination in human subjects with and without depression.

#### 3.1.8. Diet

Among the dietary factors investigated in relation to depression, there is consistent evidence of a link between the levels of sugar consumption and the prevalence of this disorder [44,134,135]. A study of healthy volunteers found that consumption of a meal with a high glycemic index was associated with a blunted release of motilin compared with a meal with a low glycemic index [136]. Similarly, the consumption of glucose in diabetic patients undergoing a glucose tolerance test resulted in a significant decrease in plasma motilin [137]. In contrast, the administration of prebiotics resulted in increases in motilin levels in an animal model of constipation [138]. Short-chain fatty acids (SCFAs), particularly those synthesized by gut bacteria, have been identified as an important mediator of gut–brain axis interactions in depression [139]. Low levels of fecal SCFAs have been observed in a mouse model of depression [140], while administration of dietary fiber increased SCFA formation by gut bacteria, which was linked to increased monoamine expression and reduced depressive-like behavior in mice [141,142]. While direct administration of SCFAs does not appear to affect the motilin levels [143], administration of a dietary symbiotic was associated with concurrent increases in the levels of motilin and gut SCFAs in an animal model of constipation, which correlated with improvements in bowel functioning [144]. These findings suggest that a significant relationship exists between certain dietary components and motilin secretion, which may merit further examination when examining dietary gut–brain axis relationships in depression.

#### 3.1.9. Antidepressants

A study of healthy volunteers receiving low-dose (37.5 mg) amitriptyline, a tricyclic antidepressant which is also effective in functional gastrointestinal disorders, found no evidence of a significant change in the motilin levels [145]. This may reflect the complex pharmacology of this agent, which includes the inhibition of serotonin and noradrenaline reuptake as well as muscarinic receptor blocking. However, this dose is substantially lower than that normally used in the treatment of depression. In contrast, in a study of patients with depression receiving tricyclic antidepressants at therapeutic doses, significant elevations in basal plasma motilin were observed [146]. Among the selective serotonin reuptake inhibitors, fluoxetine was not associated with significant changes in motilin levels or gastrointestinal motility in an animal model of chronic stress [147], and paroxetine had no significant effect on the motilin levels in human patients [73]. However, the dose of paroxetine used in this trial (20 mg/day) was at the lower end of the therapeutic range. In contrast, the newer antidepressant mirtazapine, which increases monoamine release and blocks specific serotonin receptor subtypes, appeared to increase the motilin levels in patients with a functional gastrointestinal disorder when given at full therapeutic doses (30 mg/day). This increase was associated with improvements in both gastrointestinal and depressive symptoms [74]. These findings suggest that different antidepressant classes have distinct effects on the plasma motilin levels and that investigation of the relationship between changes in the plasma motilin and antidepressant response may be warranted. Moreover, changes in the motilin levels during antidepressant therapy may be dose-dependent. However, some caution in warranted in interpreting these results, as details of concurrent medications received by these patients were not provided; it is not known if other drugs, such as anxiolytics or mood stabilizers, may have affected the motilin levels through their GABAergic actions.

It is also of note that macrolide antibiotics, which act as motilin receptor agonists, have been associated with the emergence of manic symptoms, including a report of a “switch” to mania in a patient with depression [148,149]. Though the true frequency of such effects is unclear, they may be relevant given that similar phenomena are reported during antidepressant therapy [150].

#### 3.1.10. Summary

There is evidence for direct or indirect links between motilin and several mechanisms relevant to depression and antidepressant drug responses, though much of this evidence has emerged from experimental research or single-human studies and requires verification and replication in clinical settings. These pathways are summarized in Figure 1. The mechanisms reviewed above may not exhaust all the possible links between motilin and depression. For example, there is preliminary evidence of interactions between motilin and the composition of the gut microbiota [151] and between the levels of motilin and the peptide transmitter neuropeptide Y [73,126], which has also been identified as having a protective effect against depression [152].

### 3.2. Correlations between MLN rs2281820 Allele Frequencies and the Prevalence of Depression across Countries

Data on a total of 26 countries were analyzed: Belarus, Cambodia, China, Colombia, Denmark, Estonia, Finland, France, Hungary, Ireland, Italy, Japan, Kazakhstan, Kyrgyzstan, Mexico, Moldova, Nigeria, Peru, the Russian Federation, South Korea (Republic of Korea), Spain, Ukraine, the United Kingdom, the United States of America, Uzbekistan and Vietnam. The estimated prevalence of depression in these countries ranged from a minimum of 3.4% (Cambodia) to a maximum of 6.3% (Ukraine), with a mean prevalence of 4.8 ± 0.7%. The estimated allele frequency (C allele) of the MLN rs2281820 functional polymorphism ranged from a minimum of 46.3% (Colombia) to a maximum of 94.4% (South Korea), with a mean of 63.2 ± 14.8%.

Bivariate correlations between the estimated prevalence of depression, rs2281820 C allele frequency and potential confounding variables are presented in Table 2. It was observed that there was a significant negative correlation between the MLN rs2281820 C allele frequency and the estimated prevalence of depression, as well as a positive correlation between the distance from the equator and the prevalence of depression. There were non-significant trends (0.05 < *p* < 0.1) for positive relationships between depression and both the gross national income and per capita sugar consumption. Given the high frequency of the MLN rs2281820 C allele across populations, correlations between this variable and the other confounding factors were also examined. None of these associations were statistically significant, though there was a weak trend toward a negative correlation with sugar consumption.

The following variables were included in the multivariate linear regression analysis as independent variables: MLN rs2281820 C allele frequency, distance from the equator, gross national income (all log-transformed) and per capita sugar consumption. The results of this analysis are presented in Table 3. It was found that both the rs2281820 allele frequency and distance from the equator remained significantly associated with the prevalence of depression, confirming the results of the bivariate analyses. The variance inflation factors were less than two for all variables, ruling out significant multicollinearity. This model had an adjusted R^2^ value of 0.358, indicating that it explained roughly 36% of cross-national variation in the prevalence of depression.

As a further test of this finding, stepwise linear regression was carried out using the same four independent variables. Similar results were obtained, with the rs2281820 allele frequency (β = −0.45, *p* = 0.010) and distance from the equator (β = 0.52, *p* = 0.003) included in the final model and gross national income and sugar consumption excluded. The adjusted R^2^ value for this model was 0.384, indicating slightly greater precision than for the model including all four variables.

## 4. Discussion

Gut peptides represent a promising line of research in the quest to improve our understanding of gut–brain links in common mental disorders and particularly in depression [27,28,40]. Though a systematic evaluation of the role of motilin in this process is yet to be undertaken, the existing evidence suggests that motilin may interact significantly with several key molecules and pathways that are considered to play a role in the onset and maintenance of this disorder. These include a wide range of neurotransmitters (monoamines, acetylcholine and GABA), neuroendocrine axes and, more speculatively, processes such as neural plasticity and immune-inflammatory regulation. Much of this evidence is derived from research in animal models of depression; however, the existing data on human subjects is consistent with this evidence. Because of its ability to influence afferent signaling from the gut to the brain through the vagus nerve [123] and its ability to induce central effects even when administered peripherally [127], motilin may act as a mediator in processes linking gut-related events, such as those pertaining to diet or microbiota, withr changes in brain functioning. The fact that motilin levels increase in human subjects being treated with certain groups of antidepressants in a dose-dependent manner suggests that there may be at least an indirect association between changes in the levels of this hormone and the response to these drugs [128,129]. Finally, the analysis of population-level genetic data related to a functional polymorphism of the *MLN* gene suggests that this factor may influence variations in the prevalence of depression, perhaps in combination with environmental and lifestyle factors such as exposure to stress or particular dietary practices.

Should motilin be considered a “depressant” or “antidepressant” molecule from an endogenous point of view? There is insufficient evidence to provide a definitive answer to this question. The preponderance of evidence suggests that elevated motilin levels are correlated with processes or events that are opposed to depression [72,73,97,113], while low or dysregulated levels of motilin seem to be associated with processes related to depression, such as chronic stress and unhealthy dietary practices [102,105,136]. The available evidence seems to suggest an inverse relationship between motilin and depression, though it is likely that this association will not follow a simple linear pattern. A key question for future research in this field is whether these reported associations are epiphenomenal or whether alterations in central or peripheral motilin play a more direct role in depressive disorders.

When considering the potential relationship between the motilin gene and cross-national variations in depression, it is instructive to consider the factors that are commonly cited as underlying these variations. Apart from genetic variations across populations, these factors include diet [44,134,135], socioeconomic factors such as economic inequality [54], cultural values and practices [46], altered gut microbiota [41,42] and immune-inflammatory activity caused by changes in exposure to pathogens [153]. To a greater or lesser extent, all these factors interact with gut peptides, at least in principle. Though the analysis in this paper did not find any significant association between motilin gene polymorphisms and potential confounders, this may be due to the small number of data points; analysis of data from a wider range of countries or cultures may provide leads toward some of these interactions. Socioeconomic and cultural factors are key influencers of stress, stress sensitivity and resilience [154,155], and altered patterns of motilin release in response to acute and chronic stress may influence susceptibility to depression at the individual and population levels. The possible link between diet and motilin—and in particular with sugar consumption—represents another possible area of interaction that is relevant to the development of depression. Finally, though such research is in its early stages, evidence of a link between motilin and the composition of the gut microbiome [151,156] may be particularly relevant to models that link changes in pathogen exposure and gut microbiota, such as the Pathogen Host Defense (PATHOS-D) or “Old Friends” models of depression [153,157]. Thus, variations in the release or activity of motilin, mediated through genetic variants, may interact with multiple environmental factors in influencing the risk of depression. However, given that we know little about the functional consequences of motilin gene polymorphisms and their peripheral or central effects in humans, this hypothesis should be subjected to careful testing.

Evaluating the potential of motilin as a biomarker for response to antidepressants would require consideration of several factors. First, there is evidence that some antidepressants cause an increase in motilin levels during short-term treatment. However, it is not known if these levels return to a lower or “baseline” value with more prolonged treatment. Second, though an association between changes in motilin and reductions in depressive symptoms was reported in one study, the patients in this study had a comorbid functional gastrointestinal disorder [73]. It is not clear if such an association exists in patients with depression alone or only in the sub-group of patients with depression and prominent gastrointestinal symptoms. Third, correlations between the basal motilin level or changes in this parameter (in terms of, for example, percentage changes from a baseline value) and an objectively defined response to an antidepressant (in terms of a specified percentage reduction in symptom scores) need to be examined in future studies. Finally, as shown by the negative findings with serotonin reuptake inhibitors, it is possible that this measure may be useful in assessing responses to only certain antidepressant drugs or drug groups. The effect of the antidepressant dosage, as well as the potential effects of other concurrent medications on motilin levels, should be taken into account in such studies.

Finally, when considering the therapeutic potential of drugs acting via motilin receptors in patients with depression, certain clues may be obtained from clinical trials of drugs acting on the motilin receptor in other conditions. Macrolide antibiotics, which act as agonists of the motilin receptor, are commonly used to treat disorders of gastric motility. The presence of depressive symptoms in these disorders is generally associated with a worse response to treatment in these patients [158], but it is not known if treatment with macrolides improves depressive symptoms in parallel with improvement in upper gastrointestinal symptoms. Exposure to higher or longer courses of macrolide antibiotics such as azithromycin and clarithromycin has been associated with an increased risk of depression in certain clinical scenarios [159,160]. However, it is not possible to conclude whether these effects are due to the actions of these drugs at the motilin receptor or to their effects on the intestinal microbiome. Moreover, these effects may be related the underlying medical conditions for which these antibiotics were prescribed, such as peptic ulcer disease, which is associated with depression [161,162]. There have also been numerous reports of new onset mania associated with the use of these drugs [148,149,163,164], which may suggest an association between peripheral motilin receptor activation and changes in mood. In this connection, it is also noteworthy that peripheral administration of erythromycin to healthy subjects resulted in a decrease in growth hormone levels, an action opposite to that of centrally administered motilin [165]. It is thus possible that peripheral and central motilin receptor activation may have distinct effects on one’s mood. On the other hand, azithromycin has been observed to exert antidepressant effects in a mouse model of depressive-like behavior when administered peripherally [166]. While these findings cannot be generalized directly to patients with depression, it may be possible that motilin receptor agonists have distinct actions in depressed and non-depressed individuals. Though more selective motilin agonists have been developed [167], data on their use in clinical samples is limited. Moreover, it is possible that centrally acting agonists may have more beneficial effects [113]. In contrast with these findings, we know much less about the effects of motilin receptor blocks in humans. Synthetic antagonists of the motilin receptor, such as MA-2029, have been developed for experimental use and have been found to interfere with specific phases of the gastrointestinal migrating motor complex and reduce the frequency of bowel movements in animals [168,169]. However, there are no published reports of the effects of motilin receptor blockers on mood, behavior or the levels of neurotransmitters or other hormones in humans or animals. Testing these drugs in animal models of depression would further refine our understanding of the links between motilin and mood. Though it would be premature to conduct clinical trials of existing motilin receptor agonists in patients with depression at this point in time, there are several potential lines of enquiry that follow from the existing evidence: (1) assessment of changes in depressive symptoms in patients receiving motilin agonists for an existing indication, such as diabetic gastroparesis, (2) assessment of changes in mood in patients receiving macrolide antibiotics for infectious diseases, (3) evaluation of motilin or motilin agonists and antagonists, both centrally and peripherally administered, in animal models of depression and finally (4) the development and testing of centrally acting motilin receptor agonists or antagonists in depressive or anxiety disorders. Even if such strategies do not directly lead to the approval of a specific drug, they would deepen our understanding of the role that this gut peptide and related molecules play in gut–brain “cross-talk” in depression. This could in turn lead to the development of novel therapeutic approaches aimed at targeting gut–brain axis dysfunction in this disorder.

The work presented here is subject to certain important limitations. Concerning the first stage, these include a paucity of relevant studies in patients with depression, access to a limited number of databases for the literature search, difficulties in extrapolating findings from animal models of depression, the indirect nature of much of the available evidence linking motilin to the pathophysiology of depression, the lack of replication of the available positive findings and the lack of research linking motilin to those mechanisms, particularly alterations in the microbiome, which are considered to be more relevant in depression. Moreover, more fundamental details of the physiological significance of motilin in relation to brain functioning when compared to other relevant peptides such as ghrelin and somatostatin and the interactions between them require elucidation [170]. It is also a fact that peptide molecules do not easily cross the blood–brain barrier, which limits their therapeutic potential as treatments for depression at the central level [171]. However, this difficulty could be circumvented by the development of synthetic non-peptide or small-molecule motilin receptor modulators [167,172]. In addition, in interpreting the results of studies examining the link between antidepressants and motilin levels, the possible confounding effect of an antidepressant dose or other psychotropic medications should be taken into consideration. For the associations reported in the second stage of this work, key shortcomings include a relatively small sample size, reliance on estimates of the prevalence of depression rather than direct epidemiological research, the possibility that other confounding factors may have been overlooked in the analysis, the limitations of inferring causation from population-level associations [173], and our lack of knowledge regarding the functional consequences of the *MLN* polymorphism being analyzed, particularly in terms of gut–brain axis functioning. 

## 5. Conclusions

Despite the limitations enumerated above, the possibility of a link between motilin and several key physiological processes related to depression is biologically plausible and supported by the limited evidence available to date. Though it is not possible to draw definitive conclusions from this evidence, it is likely that a more in-depth understanding of the functions of motilin in gut–brain signaling, both in healthy subjects and in patients with depression, would emerge from further human and animal studies. Evaluation of the possible central nervous system and behavioral effects of motilin agonists and antagonists would permit a better delineation of the extra-digestive functions of the motilin receptor in animals and humans. This would improve our understanding of this disorder and could lead to novel strategies—either based on direct modulation of motilin receptor activity or on indirect approaches such as dietary modification or the administration of probiotics—to normalize gut–brain functioning and improve recovery rates in patients with depression. The relationship between antidepressant use, symptomatic response and plasma motilin levels may also merit further examination, both as a marker of response and as a potential means of distinguishing responders and non-responders to drug treatment in depression.

## Figures and Tables

**Figure 1 life-11-00892-f001:**
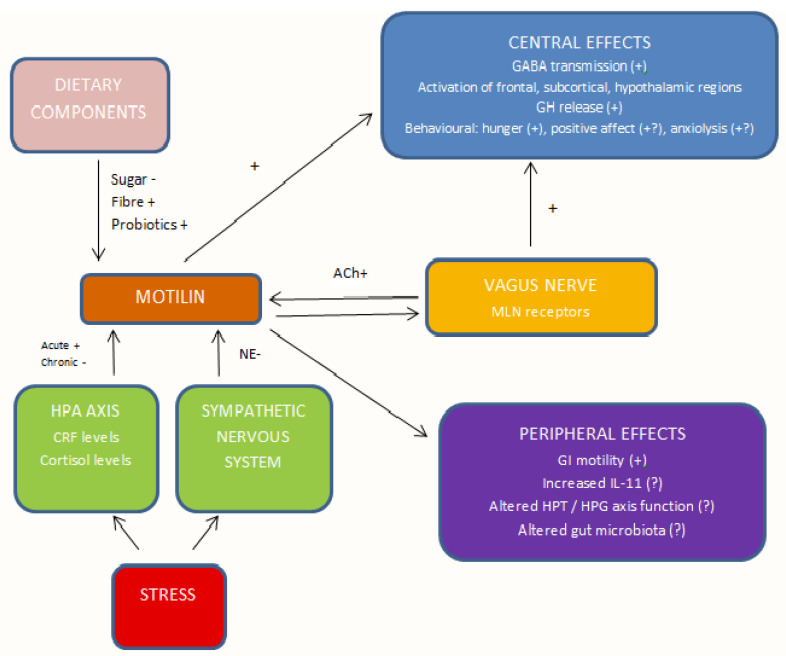
Illustration of the relationships identified between motilin and various factors or processes relevant to the pathogenesis and treatment of depression. Abbreviations: MLN, motlin; ACh, acetylcholine; NE, noradrenaline; GABA, gamma–aminobutyric acid; HPA, hypothalamic–pituitary–adrenal; CRF. corticotrophin-releasing factor; GI, gastrointestinal; IL-11, interleukin-11; HPT, hypothalamic–pituitary–thyroid; HPG, hypothalamic–pituitary–gonadal; +, stimulatory effect; -, inhibitory effect; ? indicates limited evidence for the effect.

**Table 1 life-11-00892-t001:** Search terms used for the conceptual review of links between motilin and depression.

Mechanistic Pathway and Supporting References	Search Terms Used
Neuroendocrine axes [14,26,43]	“cortisol”, “corticotropin-releasing hormone”, “corticotropin-releasing factor”, “CRH”, “CRF”, “hypothalamic-pituitary-adrenal axis”, “growth hormone”, “thyroid”, “thyroxine”, “thyroid-stimulating hormone”, “thyrotropin-releasing hormone”, “luteinzing hormone”, “follicle-stimulating hormone”, “estrogen”, “estradiol”, “progresterone”, “progestin” and “testosterone”
Stress and stress responses [26,35]	“stress”, “stressor”, “stress response”, “stress sensitivity” and “resilience”
Monoamine neurotransmitters [10]	“monoamine”, “serotonin”, “dopamine”, “noradrenaline” or “norepinephrine”, with and without “receptor *”
Other relevant neurotransmitters [15]	“gamma–aminobutyric acid”, “GABA”, “glutamate”, “acetylcholine”, “cholinergic”, “neuropeptide” and “neuropeptides” with and without “receptor *”
Immune and inflammatory pathways [13]	“immune”, “inflammation”, “inflammatory”, “cytokine *” and “chemokine *”
Neurotrophic factors [12]	“brain-derived neurotrophic factor”, “BDNF”, “neural plasticity” and “neuroplasticity”
Diet [44]	“diet *”, “sugar”, “refined sugar”, “probiotic*”, “prebiotic *”, “short-chain fatty acids” and “SCFA”
Studies of antidepressants [25]	“antidepressant *” paired with “tricyclic”, “serotonin reuptake inhibitor” and “selective serotonin reuptake inhibitor”

**Table 2 life-11-00892-t002:** Bivariate correlations between MLN rs2281820 allele frequency, estimated prevalence of depression and other confounding factors.

Variable	1 Depression, Prevalence	2 MLN rs2281820, C Allele Frequency (ln)	3 Gross National Income (ln)	4 Individualism–collectivism (ln)	5 Distance from the Equator (ln)	6 Per Capita Sugar Consumption
1	-	−0.41 * (0.037)	0.38 (0.053)	0.34 (0.116)	0.49 * (0.012)	0.38 (0.053)
2		-	−0.19 (0.365)	−0.28 (0.190)	0.07 (0.754)	−0.34 (0.091)
3			-	0.75 * (<0.001)	0.47 * (0.016)	0.53 * (0.005)
4				-	0.56 * (0.005)	0.36 (0.091)
5					-	0.15 (0.452)

Abbreviations: ln, natural logarithmic transformation; MLN, motilin gene. * denotes significance at *p* < 0.05.

**Table 3 life-11-00892-t003:** Multivariate linear regression analysis of variables associated with the prevalence of depression.

Variable	Regression Coefficient (β)	Significance Level	Part Correlation	Variance Inflation Factor
MLN rs2281820, C allele frequency (ln)	−0.38	−0.037 *	−0.36	1.15
Distance from the equator (ln)	0.48	0.015 *	0.42	1.33
Gross national income (ln)	−0.01	0.950	−0.01	1.78
Per capita sugar consumption	0.19	0.359	0.15	1.53

Abbreviations: MLN, motilin gene; ln, natural logarithmic transformation. * indicates statistical significance at *p* < 0.05.

## Data Availability

The data used for the analyses in this study are available to the public, and its sources have been cited in the text. A complete data sheet is available from the author on reasonable request.

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
