# Peer review of "Gut Hormones as Potential Therapeutic Targets or Biomarkers of Response in Depression: The Case of Motilin"

_life, 2021, doi:10.3390/life11090892_

Round 1
Reviewer 1 Report
The manuscript submitted by Ravi Philip Rajkumar provides a comprehensive view of the potential role of gut hormones as potential treatments or predictors of response in depression in particular the peptide hormone motilin and the relationship between a functional polymorphism (rs2281820) of the motilin (MLN) gene and cross-national variations in the prevalence of depression. The manuscript is well-structured and great to read. I think that this manuscript is of significant interest to the field. Please find some minor comments below. Minor comments -Recheck the abbreviations, some are write in full , which makes it difficult to understand what is meant. - The Figure 1. is really simple. If it is possible, could you change it with an another more detailed figure? -Between the possible mechanisms links between motilin and depression, would be good and important to cite also the role of SCFAs, please consider adding it to the table 1 and in results.Author Response
Please see the attachment.

Reviewer 2 Report
This article deals with an important subject of great clinical relevance. About 70% of depressed patients respond to antidepressant drugs, at some level, with about 1/3 that have to be treated with other therapies, associated or not with antidepressant drugs. Thus, drugs with new mechanisms of action are very much needed, as the existing ones are based on the amine theory. Another drug that does not conform to this theory, agomelatine, which acts on melatonin receptors, has however mild to moderate efficacy. So, the pathway suggested by this author seems promising.
It has long been known that there is an embryonic and functional link between the gut and the brain, and therefore motilin may be involved in depression.
This article is a descriptive review and would be better if it were a systematic review. There must be a justification for this option.
The article is formally correct.
English should be improved, although it is acceptable (eg, line or row 54: should be has been accumulated and not has accumulated).
In the abstract there must be a description of the methodology used.
In the Introduction (rows or lines 44 to 48) there should be mention of melatonin, as there are currently antidepressant drugs that modify the action of this hormone.
In the Introduction and Discussion, there should be mention of the difficulty of peptides (eg, motilin has 22 amino acids) to cross the blood-brain barrier (BBB), which is an important therapeutic limitation.
It is mentioned in the Methodology that the author searched PubMed and Scopus databases (line 103). What is the justification for not investigating other databases?
There should be a description of the criteria used to choose the articles (it is not enough to say “possible relevance to depression” - line 112).
The Introduction, Results and Discussion did not take into account the interference in humans of drugs which can alter motilin levels and the response of motilin receptors. In humans, drugs are frequent confounding factors or bias.
There should also be a greater reference to motilin receptors. Several times the author writes about macrolides, motilin agonists, but he does not mention motilin antagonists (this could be a way to better prove the motilin pathway).
It makes correct approaches to “potential confounding facts” in the different sections of the article. He makes correct considerations about the limitations of this study.
He writes about motilin and the sympathetic system but only slightly addresses the parasympathetic / cholinergic system, the latter being predominant in the digestive tract. This relationship between motilin and the colinergic pathway must be developed.
The considerations about motilin and hyperthyroidism and hypothyroidism are confused (line 231).
In “d) monoamine transmitters” and “e” there is no mention of acetylcholine and cholinergic mechanisms, which must be done.
The correlation between the antidepressant drug groups, motilin and depression should be further developed. For example, is 37.5 mg of amitryptiline effective? (too low) How do tricyclic antidepressant drugs significantly change motilin levels (in blood) and centrally? But selective serotonin reuptake inhibitors (SSRI) do not change them. There must be an explanation. Also here, the issue of treatment duration should be addressed because antidepressants are effective only after 2-4 weeks, but the influence on motilin levels is rapid. And with the SSRI?
It is also not correct to state that macrolide antibiotics can cause manic symptoms and that this is an indication of a possible antidepressant effect (line 389). This statement should be further clarified. We also have doubts about the correlation between the use of macrolide antibiotics and the frequency of depression (line or row 523). For example, clarithromycin can be given for 10 days to two weeks to eradicate Helicobacter pylori and there appears to be no more depression. On the contrary.
The estimated allele frequency of the MLN rs228182o is very high (46.3-94.4%) (line or row 416) which means that it is correlated with a lot of phenomena, including depression and motilin. Some clarification is needed (at least in the Discussion).
There should often be more caution in some statements, opting for “may be” and not being totally affirmative, for example, “to play a role” (line 453). This happens many times.
In the Discussion there should be more details about motilin receptors because this could be the future way of acting in depression and other diseases.
There is an extensive and correct bibliography.
In conclusion, it is a good review that deserves to be published once these issues have been clarified.
